# Relation between occupants' health problems, demographic and indoor environment subjective evaluations: A cross-sectional questionnaire survey study in Java Island, Indonesia

**Solli Murtyas**[1]*, **Nishat T. Toosty**[1,2], **Aya Hagishima**[1], **N. H. Kusumaningdyah**[3]

**1** Department of Energy and Environmental Engineering, Interdisciplinary Graduate School of Engineering Sciences, Kyushu University, Fukuoka, Japan, **2** Department of Statistics, University of Dhaka, Dhaka, Bangladesh, **3** Department of Architecture, Faculty of Engineering, Universitas Sebelas Maret, Surakarta, Indonesia

* s.murtyas@kyudai.jp

**Data Availability Statement:** All relevant data are within the paper and its Supporting information files.

## Abstract

This study aimed to evaluate the link between health problems, demographic factors, and the indoor environment quality of residents in Indonesia. We conducted a cross-sectional design study through a questionnaire survey with 443 respondents aged between 12 and 81 years. The questionnaire was concerned with previous health problem occurrences associated with thermal discomfort experiences, indoor environments, economic conditions, and basic anthropometric factors. Logistic regression with the odds ratio (OR) was applied to evaluate the tendency of different respondent groups to suffer from certain health problems, when compared to reference groups. Furthermore, structural equation modelling (SEM) was used to incorporate certain factors (economic conditions, thermal discomfort experiences, and perceived indoor environments) into a single model to understand their direct and indirect effects on health conditions. The results indicate that economic conditions are the most significantly associated with health problems. Furthermore, we found that the low-income group was the most vulnerable to health problems, including coughing, puking, diarrhoea, odynophagia, headaches, fatigue, rheumatism, fidgeting, skin rashes, muscle cramps, and insomnia (OR: 1.94–6.04, p <0.05). Additionally, the SEM suggested that the respondents' economic conditions and thermal discomfort experiences had significant direct effects on their health problems with standardized estimates of -0.29 and 0.55, respectively. Additionally, perceived indoor environment quality, which is possible to cause thermal discomfort experience, indirectly affect health problems. These findings contribute an insightful and intuitive knowledge base which can aid health assessments associated with demographic and physical environments in developing sustainable and healthy environment strategies for the future.

**Funding:** This study was financially supported by Obaysahi Foundation, Japan. Funder: Obayashi Foundation Award Number: Not available Website: obayashifoundation.org Grant Recipient: Aya Hagishima The funders had no role in study design, data collection and analysis, decision to publish, or preparation of the manuscript.

**Competing interests:** The authors have declared that no competing interests exist.

# Introduction

The current state of urban dwellings in the world is a major issue that needs to be addressed in order to achieve sustainable development goals (SDGs); with a particular focus on good health and well-being [1]. In developing countries, housing conditions have strong correlations with the Human Development Index [2]. Therefore, improving the quality of housing in these countries is a key factor in enhancing a decent standard of living, long and healthy life, and education [3].

Indonesia, the fourth most populous country in the world, had an estimated population of over 270 million in 2019, which is projected to be around 300 million by 2030. Currently, 9.2 percent of the population lives below the national poverty line [4] and the country is facing serious constraints in providing adequate housing [5, 6]. The lack of quality and affordable housing among 41% of the informal settlements due to insufficient financing became apparent in the last decade [7]. Moreover, the public health risks posed by contaminated water, malnutrition, air pollution, and inadequate environments resulted in higher mortality and morbidity rates [8]. Therefore, the increasing health problems related to the country's quality of living have become more epidemiologically complex in recent years [9].

In response to the health-related issues caused by certain climates and environments, the World Health Assembly (WHA)–the decision-making body of the World Health Organization (WHO)–has adopted the International Health Regulations (2005) in order to encourage countries to establish definitive health policies through strategic health detection, assessments, reports, and management as part of mitigating public health risks [10]. However, the unavailability of evidence-based data and knowledge of how the physical environment relates to individual factors has made this task difficult [11]. Consequently, previous studies have mainly focused on the impact of the environment on health outcomes. Gao et al. (2016) pointed out that built environments are linked to lifestyles; therefore, they can contribute to an increased risk of chronic health problems [12]. Based on an epidemiological investigation in China, Xiong et al. (2017) found an association between climate, housing environment, and air-conditioning usage with occupant's discomforts, including physiological and nervous system effects [13]. Moreover, Yang et al. (2020) conducted structural equation modelling (SEM) in a survey of 591 respondents aged between 18 and 68 years in 2017 to understand the pathway effects between perceived residential green spaces and mental health in China [14]. They found a significant indirect effect of perceived green spaces on mental health, specifically with regard to environmental disturbances and social cohesion. In Indonesia, Jong et al. (2018) adopted the epidemiological approach reported by WHO and the Ministry of Health of the Republic of Indonesia; they found that due to environmental degradation and the impact of climate change, the general public's health is prone to vector-borne and zoonotic diseases [15].

Therefore, understanding the basic relationship structure between the built environment, anthropometric factors, and economic conditions for a specific region is vital. Hence, this study intends to reveal the relationship between perceived indoor environments, experiences of indoor thermal discomfort, and occupant health conditions in Indonesia by means of a subjective questionnaire survey. Furthermore, the factors related to indoor environment quality (IEQ), economic levels, locations, and basic anthropometric information are also considered as potential explanatory variables for health conditions.

## Methods

### Questionnaire design

This study implemented a cross-sectional study design with a questionnaire survey. Fig 1 displays the information collected during the investigation. The questionnaire is comprised of 72

| Personal factors |
|---|
| • Age |
| • Gender |
| • Location |
| • Economic level |

| Indoor environment conditions |
|---|
| • Indoor air quality satisfaction |
| • Subjective indoor thermal evaluation |
| • General indoor environments satisfaction |

**Health problems**

*Respiratory*
- Sinusitis
- Bronchitis
- Cough
- Tuberculosis
- Pneumonia
- Asthma
- Dyspnea

*Kidney and urinating*
- Urinary hesitancy
- Inflammation
- Prostate inflammation
- Urolithiasis

*Neural*
- Meningitis
- Concussion
- Polio
- Epilepsy
- Stroke
- Migraine

*Digestive tract*
- Typhoid
- Puke
- Constipation
- Ulcer
- Icterus
- Cholelithiasis
- Hemorrhoids
- Diarrhea
- Odynophagia

*Gland*
- Thyroid

*Dermal*
- Chicken pox
- Tinea versicolor
- Eczema

*Heart and blood*
- Tachycardia
- Chest pain
- Hypotension

- Hypertension

*Bodily pains*
- Headache
- Fainting
- Tiredness
- Rheumatism
- Skin rash

*Mental*
- Insomnia
- Fidgeting

*Other*
- Severe dehydration
- Medicine allergy
- Food allergy
- Tetanus
- Cancer
- Malaria
- Diabetes
- Measles
- Hearing disorder

**Fig 1. The information collected from the questionnaire survey.**

questions concerned with demographic conditions, indoor environment conditions, and the self-reported health problems recorded in the last three months. The demographic conditions included: age, gender, location of residence, and economic conditions.

Several questions pertained to occupants' subjective evaluation of indoor air quality, thermal discomfort experiences, and indoor environment conditions. In the questions concerned with their subjective feelings of the indoor air quality, respondents evaluated their satisfaction levels as one of four ratings, namely "very good", "good", "slightly poor", and "poor". Furthermore, if they gave a negative evaluation of the indoor air quality (i.e. "slightly poor" and "poor"), they were asked to comment on the potential sources of the poor indoor air quality.

The subjective evaluations concerning indoor thermal conditions were measured with the question, "How would you rate the general indoor thermal conditions?" Their answers were limited to a five-point scale divided into "cold", "cool", "neutral", "warm", and "hot".

Since Indonesia is located on the equator and has a tropical climate, the mean outdoor temperature is relatively high; ranging from 26°C to 31°C throughout the year [16]. Meanwhile, the household penetration rate of air conditioners (ACs) was reported to be only 9 percent in 2015 [17]. Furthermore, the insulation performance of residential buildings, especially for low-income households in this country, is notoriously poor. Murtyas et al. (2020) pointed out that people living in low-cost housing with inadequate insulation materials were susceptible to temperatures over 30°C for 5 to 12 hours a day [18]. In light of these findings, we used the following questions to examine the respondents' thermal discomfort experiences: "How often did you struggle to sleep because of uncomfortable temperatures?" and "How often were you exposed to hot indoor air temperatures?"; respondents were asked to select one of the available answers: "Never", "Less frequent", "A few days or nights per month", "A few days or nights per week", and "Every day". Additionally, indoor environment conditions were evaluated by asking respondents to indicate their level of satisfaction with respect to the holistic conditions of their indoor living spaces; the relevant responses were: "Very satisfied", "Satisfied", "Slightly dissatisfied", and "Dissatisfied".

The questions related to self-reported health problems were in accordance with the regulations stipulated by the Ministry of Health of Indonesia No. 29, 2013 [19]. This survey is concerned with major health conditions related to respiratory, kidney, urinary, neural, digestive tract, gland, dermal, heart and blood, bodily pains, mental health, and other health problems. The complete questionnaire form is presented in S1 Appendix. The questionnaire survey in this study was anonymous, and participants decided whether or not to participate after receiving sufficient explanation about the research objective. Obtaining a signed consent from participants were waived. All procedures performed in this study were in accordance with the ethical standards of Kyushu University and with the 1964 Helsinki declaration and its later amendments or comparable ethical standards, being approved by the Ethical Committee of Kyushu University.

## Data collection

This study included 443 respondents who were selected from the six provinces of Java Island, namely Banten, Jakarta, West Java, Central Java, Yogyakarta, and East Java—as shown in Fig 2. These provinces represent an area with higher population density than other Indonesian provinces. The data were obtained through an online survey and direct interviews from December 2019 to January 2020 (typical wet season); before the first case of COVID-19 was found in Indonesia. In the online survey section, 306 respondents were asked to fill out the online form. The surveyors targeted respondents on social media that fit the inclusion criteria, such as aged between 12 and 70 years old and living in either rural or urban districts located in the above-mentioned provinces. Furthermore, in order to understand the real health and living conditions in their environments, we also conducted a direct questionnaire survey with 137 respondents by visiting them door-to-door in Kampung Sangkrah, Surakarta City, Central Java Province—which is identified as an urban kampung district [20]. The respondents who were included from this district were between 12 and 70 years old and have been living in these districts for more than five years with legal and permanent residencies.

Table 1 displays the breakdown of the respondents' personal characteristics. Over half of the respondents were female (59.8%), while only 40.2% were male. The respondents' locations are divided into urban, rural, and urban kampung—terminology from Indonesian language which has similar meaning with urban slum—districts, with proportions of 38.2%, 30.9%, and

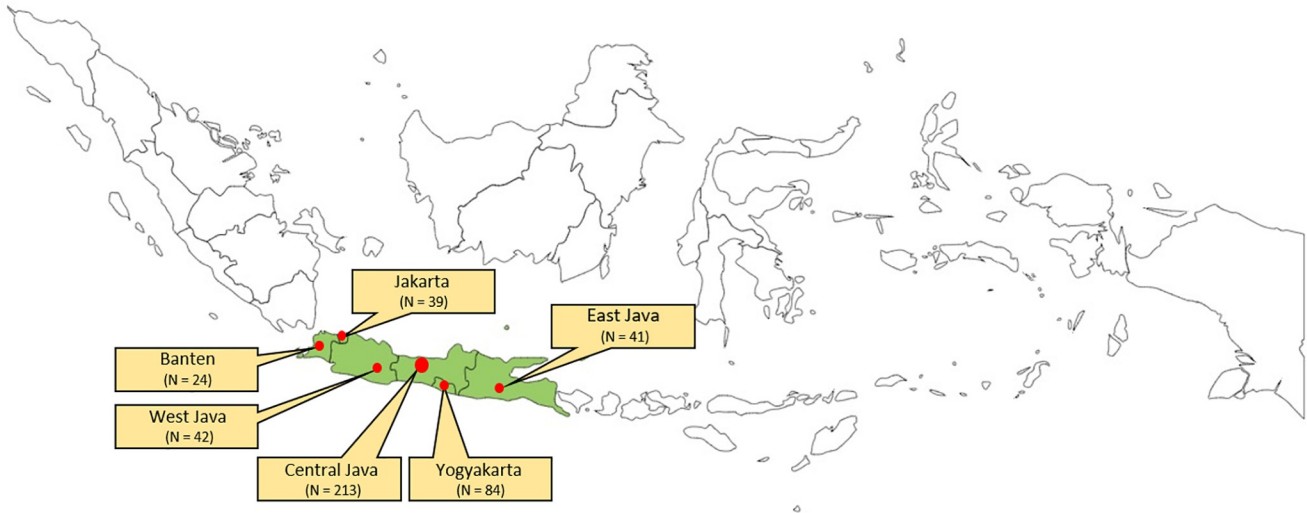

**Fig 2. The distribution of respondents who completed the questionnaire survey from the six provinces on Java Island.**

30.9%, respectively. Approximately three-quarters of the respondents spent more than thirteen hours a day at home. The majority of respondents' economic conditions were categorized as low income (30.5%).

## Results

### Subjective evaluations of indoor thermals and general indoor environments

Fig 3 shows the frequencies of the respondents' different subjective experiences related to indoor thermal discomfort as classified by their age, economic level, and satisfaction with the

Table 1. Characteristics of the respondents (N = 443).

| Respondent characteristics | Total number (N) | Percentage |
|---|---|---|
| Group of age | | |
| <18 years old | 63 | 14.2% |
| 18–30 years old | 228 | 51.5% |
| 31–40 years old | 87 | 19.6% |
| 41–50 years old | 29 | 6.5% |
| >50 years old | 36 | 8.2% |
| Gender | | |
| Male | 178 | 40.2% |
| Female | 265 | 59.8% |
| Locations | | |
| Urban | 169 | 38.2% |
| Rural | 137 | 30.9% |
| Urban kampung | 137 | 30.9% |
| Income level | | |
| Lower | 135 | 30.5% |
| Lower-middle | 120 | 27.1% |
| Middle-upper | 72 | 16.2% |
| High | 116 | 26.2% |

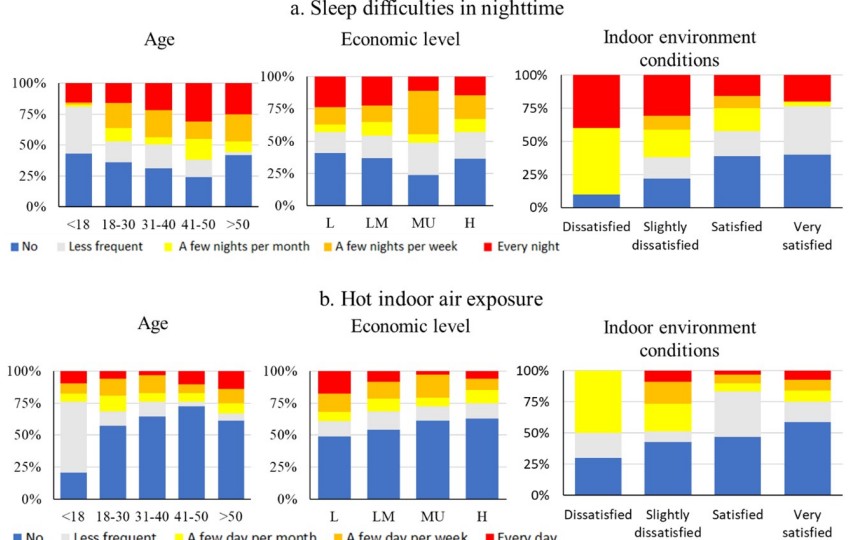

**Fig 3. Percentage of respondents who experienced sleep difficulties and hot indoor air exposure according to their age, economical level, and indoor environmental conditions.**

general indoor environment. The ratio of respondents who experienced sleep difficulties more than a few days per week increased from 21% to 44% as their ages increased, except for the group aged over 50 years old (see Fig 3a). This trend between different age groups is generally consistent with previous studies, which reported that sleep difficulties are more common in older adults than in younger age groups [21, 22]. Furthermore, this result is also in line with previous studies concerned with the physiological factors and ages associated with the risk factors of heat stress [23]. Compared to younger generations, the elderly exhibit reduced thermoregulatory responses related to their sweating rate, skin blood flow, and cardiovascular functions [24].

Conversely, the data representing respondents from different age groups who felt hot due to high indoor air temperatures revealed a different tendency compared to those with difficulties in sleeping. The ratio of people who have experienced hot indoor temperatures was much smaller in the age groups below 18 years, when compared to the older age groups. Another study emphasized a similar finding which asserts that the body surface area per unit body-weight is greater in children and young people than in adults. This makes it more difficult for children to reserve water in their bodies, especially under heat stress conditions [25].

Additionally, the observed data revealed a notable trend regarding the different frequencies of sleeping difficulty and feeling high indoor temperatures between different income groups. The authors expected that the higher income groups would be inclined to live in more expensive houses with better indoor thermal environments; as a result, the frequency of both a difficulty to sleep and the experience of high temperatures were predicted to be lower than in the other income groups. However, the observed ratio of respondents who did experience these difficulties more than a few days per week increased in the low, lower-middle, and middle-income groups, in that order. Likewise, the ratio among the high-income groups exhibited levels similar, and even slightly higher, than the low-income group. The reason for this discrepancy between our previous assumptions and the observed data might be due to the different acceptable living standards and ratios between white-collar workers and manual labourers in different income groups. Contrastingly, the data related to feeling high indoor air

temperatures between the income groups revealed a tendency opposite to the data of struggling to sleep. The ratio of frequently experiencing hot conditions increased from the high-income group to the low-income group. Hot conditions are generally not only during high atmospheric temperatures but also when the body undergoes strenuous physical activities, which causes intensive sweating [26]. This likely occurs in the more susceptible low-income group, which is supposed to have a high percentage of manual labourers working outdoors.

Furthermore, based on respondents' subjective evaluations, those who indicated that they were "very satisfied" with indoor environment conditions, generally had a small fraction of having difficulties to sleep in nighttime and suffered hot indoor air exposure. The opposite was true for the respondents who indicated that they were "dissatisfied" with their indoor environment conditions. This finding implies that inadequate indoor environmental conditions could possibly undermine sleep quality. This is in line with a study by Urlaub et al. (2015), who emphasized the importance of physical environments (thermal conditions and indoor air quality) in obtaining good sleep [27].

## Subjective evaluation of indoor air quality

Fig 4 displays the respondents' subjective evaluations of the air quality in their places of residence. The respondents living in rural districts indicated the highest percentage of satisfaction with their indoor air quality (82.5%). In contrast, the urban kampung group showed the lowest percentage of satisfaction (73.7%). Furthermore, we found that dust was considered the main source of poor air quality by 58%, 66%, and 41% of the respondents in urban, rural, and urban kampung districts, respectively. Interestingly, the number of rural respondents who suggested that dust pollutants are the main cause of air pollution were higher than those in the urban and urban kampung districts. Santoso et al. (2020) conducted a quantitative measurement of airborne particulate matter (PM) and black carbon content; they revealed that the average annual $PM_{2.5}$ concentrations between 2010 and 2017 in most of the urban districts of Java Island (Jakarta, Bandung, Surabaya, and Yogyakarta) were higher than the Indonesian annual ambient air quality standard (15 $\mu g.m^{-3}$). Conversely, the rural areas observed in Pekanbaru

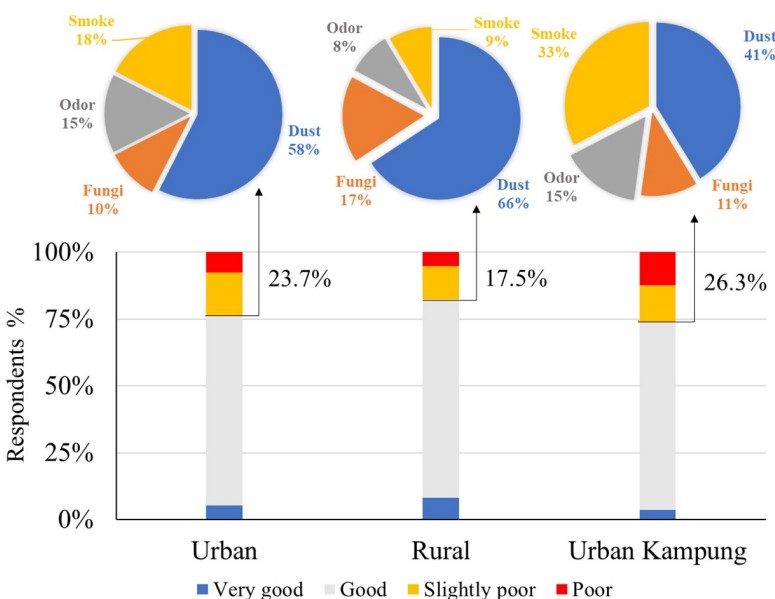

**Fig 4. A subjective indoor air quality evaluation according to respondents from different locations.**

and Palangka Raya were lower than the national standard [28]. These observations are consistent with the respondents' perceptions in our survey.

## Logistic regression analysis of the prevalence of health problems

Table 2 displays the questionnaire results pertaining to the prevalence of health problems. This information enabled us to examine the relationship between respondents' health problems and other variables based on logistic regression with an odds ratio (OR). The OR measures the effect of a predictor on the likelihood that the outcome of interest would occur. In our study, perceived indoor environments, lifestyles, personal factors, and dwelling conditions were designated as independent variables, and then their influence on health problems was evaluated. Hence, an OR less than 1 implies a negative relationship and vice versa [13, 29, 30].

$$Logit(\theta) = \ln\frac{\theta(Y=1)}{\theta(Y=0)} = \beta_0 + \beta_1 x_1 + \beta_2 x_2 + \cdots + \beta_n x_n \tag{1}$$

The logistic regression is expressed in Eq (1). where $\theta$ (Y = 1) is the probability of the respondents' suffering from a certain symptom, $\beta_o$ is a constant, $x_j$ (j = 1,2,..., n) is the predictor variable, and $\beta_j$ (j = 1,2,..., n) is the coefficient of the corresponding predictor variable. The OR is obtained by taking the exponent of the estimated coefficients along with the 95% confidence interval. Statistical analysis was performed using R software version 3.6.3 with the generalized linear model package.

We hypothesized that the respondents' personal variables (age, gender, and economic conditions) are basic driving factors that affect health conditions. Hence, we initially conducted an analysis of the logistic regression relating to personal factors and health problems. The relationships between health problems and predictors that were evaluated as being statistically significant, at levels below 5%, are shown in Table 3. Furthermore, a negative value of the estimated $\beta_j$ coefficients (OR<1) indicates a negative association between the prevalence of the health problem and the corresponding covariate. It means that respondents from a certain group of the corresponding covariate are less prone to the specific health problem compared to those in the reference category. Similarly, OR>1, which occurs for a positive value of $\beta_j$ estimates, indicates higher vulnerability of that specific group of the respondents to the health problem of interest than those of the reference group.

Consequently, with regard to age, the group younger than 18 years old was found to be more vulnerable to several significant mental health problems (e.g. fidgeting and insomnia), bodily pains (e.g. headache), and digestive tract issues (e.g. constipation) when compared to a reference group, such as the one older than 50 years. The trend pertaining to a high prevalence of mental health problems among younger ages—of which fidgeting was the highest (OR: 4.74)–is supported by the study on mental health problems in school-aged children in Aceh Province, Indonesia [31]. They pointed out that 37.8% of the observed school-aged children exhibited negative emotional symptoms originating from maternal parenting stress. Furthermore, the national data indicates that 10% of Indonesian adolescents exhibited poor mental health; 32% of which had experienced bullying at school [31, 32]. With regard to hypertension, it has been long known that interactions of multiple genetic and environmental factors play a significant role, and identified several indices associated with prevalence of hypertension such as body mass index [33–35]. Because of the limitation of sample number and type of questions, the survey cannot draw insight related to these factors, we can confirm that the prevalence of hypertension tends to be significantly correlated with the age group older than 50 years since its odds ratio is less than one; the lowest of all the age groups. Although the sample number of our survey is not quite large, this tendency is consistent with the Indonesian Ministry of

**Table 2. A summary table of respondents' past health problem experiences.**

| Health problems | | N | % |
|---|---|---|---|
| Respiratory | Cough | 238 | 53.7% |
| | Sinusitis | 43 | 9.7% |
| | Dyspnea | 39 | 8.8% |
| | Bronchitis | 33 | 7.4% |
| | Asthma | 24 | 5.4% |
| | Tuberculosis | 0 | 0.0% |
| | Pneumonia | 0 | 0.0% |
| Kidney and urinating | Urinary hesitancy | 15 | 3.4% |
| | Inflammation | 15 | 3.4% |
| | Prostate inflammation | 0 | 0.0% |
| | Urolithiasis | 0 | 0.0% |
| Neural | Migraine | 125 | 28.2% |
| | Meningitis | 0 | 0.0% |
| | Concussion | 0 | 0.0% |
| | Polio | 0 | 0.0% |
| | Epilepsy | 0 | 0.0% |
| | Stroke | 0 | 0.0% |
| Digestive tract | Ulcer | 149 | 33.6% |
| | Diarrhoea | 127 | 28.7% |
| | Constipation | 108 | 24.4% |
| | Puke | 97 | 21.9% |
| | Odynophagia | 42 | 9.5% |
| | Typhoid | 34 | 7.7% |
| | Haemorrhoids | 27 | 6.1% |
| | Icterus | 0 | 0.0% |
| | Cholelithiasis | 0 | 0.0% |
| Gland | Thyroid | 14 | 3.2% |
| Dermal | Tinea versicolor | 44 | 9.9% |
| | Eczema | 39 | 8.8% |
| | Chicken pox | 23 | 5.2% |
| Heart and blood | Hypotension | 97 | 21.9% |
| | Tachycardia | 48 | 10.8% |
| | Hypertension | 44 | 9.9% |
| | Chest pain | 42 | 9.5% |
| Other | Fatigue | 266 | 60.0% |
| | Headache | 226 | 51.0% |
| | Muscle cramps | 218 | 49.2% |
| | Insomnia | 111 | 25.1% |
| | Severe dehydration | 99 | 22.3% |
| | Fidgeting | 99 | 22.3% |
| | Rheumatism | 89 | 20.1% |
| | Skin rash | 47 | 10.6% |
| | Fainting | 18 | 4.1% |
| | Measles | 10 | 2.3% |
| | Food allergy | 0 | 0.0% |
| | Medicine allergy | 0 | 0.0% |
| | Hearing disorder | 0 | 0.0% |
| | Tetanus | 0 | 0.0% |
| | Cancer | 0 | 0.0% |
| | Malaria | 0 | 0.0% |

**Table 3. Logistic regression and odd ratio relating personal factors to health problems.**

| Personal factors | Health problems | Estimate | p-value | SE | OR | OR with 95% CI | |
|---|---|---|---|---|---|---|---|
| | | | | | | Lower limit | Upper limit |
| *Gender (female as a reference)* | | | | | | | |
| Male | Tinea versicolor | 1.07 | 0.00123 ** | 0.33 | 2.90 | 1.54 | 5.66 |
| | Headache | -0.48 | 0.0133 * | 0.1953 | 0.62 | 0.42 | 0.90 |
| | Migraine | -0.59 | 0.00891 ** | 0.2249 | 0.56 | 0.35 | 0.86 |
| | Fatigue | -0.62 | 0.00176 ** | 0.1983 | 0.54 | 0.36 | 0.79 |
| | Muscle cramps | -0.79 | 0.00007 *** | 0.19831 | 0.46 | 0.31 | 0.67 |
| | Constipation | -0.78 | 0.00134 ** | 0.2434 | 0.46 | 0.28 | 0.73 |
| | Hypotension | -0.96 | 0.00025 *** | 0.2621 | 0.38 | 0.23 | 0.63 |
| | Ulcer | -1.03 | 0.0001 *** | 0.22267 | 0.36 | 0.23 | 0.55 |
| *Age groups (> 50 as a reference)* | | | | | | | |
| < 18 | Fidgeting | 1.56 | 0.0007 *** | 0.4552 | 4.74 | 2.00 | 12.10 |
| | Headache | 1.55 | 0.00122 ** | 0.47819 | 4.70 | 1.88 | 12.46 |
| | Constipation | 1.19 | 0.010839 * | 0.4686 | 3.30 | 1.36 | 8.70 |
| | Insomnia | 1.12 | 0.00796 ** | 0.422 | 3.07 | 2.07 | 7.18 |
| | Migraine | -1.20 | 0.0233 * | 0.52775 | 0.30 | 0.1 | 0.83 |
| | Rheumatism | -1.32 | 0.004736 ** | 0.4684 | 0.27 | 0.10 | 0.66 |
| | Hypotension | -2.36 | 0.0317 * | 0.2304 | 0.09 | 0.00 | 0.58 |
| | Hypertension | -3.00 | 0.00517 ** | 0.14 | 0.05 | 0.00 | 0.28 |
| 18–30 | Muscle cramps | -0.87 | 0.0137 * | 0.35145 | 0.42 | 0.21 | 0.83 |
| | Hypertension | -1.13 | 0.00844 ** | 0.4299 | 0.32 | 0.14 | 0.77 |
| | Rheumatism | -1.24 | 0.0007 *** | 0.3636 | 0.29 | 0.14 | 0.60 |
| 31–40 | Muscle cramps | -0.95 | 0.0156 * | 0.39106 | 0.39 | 0.18 | 0.83 |
| | Rheumatism | -1.22 | 0.004026 ** | 0.4255 | 0.29 | 0.13 | 0.67 |
| | Hypertension | -1.47 | 0.00836 ** | 0.5579 | 0.23 | 0.07 | 0.67 |
| 41–50 | Insomnia | -1.39 | 0.04546 * | 0.696 | 0.25 | 0.05 | 0.88 |
| *Economic levels ("High" as a reference)* | | | | | | | |
| Low | Fidgeting | 1.80 | 0.0003 *** | 0.3761 | 6.04 | 3.00 | 13.29 |
| | Odynophagia | 1.64 | 0.00348 ** | 0.5615 | 5.16 | 1.89 | 18.1 |
| | Puke | 1.54 | 0.0004 *** | 0.3441 | 4.66 | 2.44 | 9.47 |
| | Insomnia | 1.53 | 0.0004 *** | 0.3275 | 4.63 | 2.49 | 9.05 |
| | Skin rash | 1.39 | 0.01461 * | 0.5712 | 4.03 | 1.44 | 14.35 |
| | Headache | 1.36 | 0.0003 *** | 0.26805 | 3.91 | 2.33 | 6.67 |
| | Muscle cramp | 1.13 | 0.00002 *** | 0.2636 | 3.11 | 1.87 | 5.25 |
| | Rheumatism | 1.11 | 0.00221 ** | 0.3626 | 3.03 | 1.53 | 6.40 |
| | Diarrhoea | 0.93 | 0.00148 ** | 0.29242 | 2.53 | 1.44 | 4.56 |
| | Cough | 0.80 | 0.00206 ** | 0.2596 | 2.23 | 1.34 | 3.72 |
| | Fatigue | 0.66 | 0.01026 * | 0.25856 | 1.94 | 1.17 | 3.24 |
| | Sinusitis | -1.24 | 0.0129 * | 0.496864 | 0.29 | 0.1 | 0.73 |

*(Continued)*

**Table 3.** (Continued)

| Personal factors | Health problems | Estimate | p-value | SE | OR | OR with 95% CI | |
|---|---|---|---|---|---|---|---|
| | | | | | | Lower limit | Upper limit |
| Low-middle | Skin rash | 1.72 | 0.00228 ** | 0.5647 | 5.60 | 2.04 | 19.75 |
| | Odynophagia | 1.22 | 0.03718 * | 0.5875 | 3.40 | 1.16 | 12.37 |
| | Fidgeting | 1.17 | 0.00302 ** | 0.395 | 3.23 | 1.53 | 7.31 |
| | Rheumatism | 1.06 | 0.00421 ** | 0.3707 | 2.89 | 1.43 | 6.18 |
| | Fatigue | 1.04 | 0.0002 *** | 0.27626 | 2.82 | 1.65 | 4.89 |
| | Headache | 0.84 | 0.00172 ** | 0.26743 | 2.31 | 1.37 | 3.93 |
| | Puke | 0.83 | 0.0231 * | 0.3666 | 2.30 | 1.14 | 4.85 |
| | Insomnia | 0.81 | 0.0201 * | 0.4288 | 2.24 | 1.15 | 4.54 |
| | Diarrhoea | 0.63 | 0.0392 * | 0.30454 | 1.87 | 1.04 | 3.44 |
| | Muscle cramp | 0.60 | 0.02348 * | 0.2666 | 1.83 | 1.09 | 3.10 |

*p < 0.05;

**p < 0.01;

***p < 0.001 by using Chi-squared test;

SE: Standard Error.

Health's report, which confirmed that 41% of adults older than 60 years old exhibited the hypertension in 2012, which is higher than the value of 17.1% for the age group of less than 25 years old [36, 37].

With respect to gender, this analysis indicates that males are more vulnerable than females to dermal problems, such as tinea versicolor. This is in line with the findings of Zara and Yasir (2019), who analyzed the effects of personal hygiene and physical environments on tinea in the fishery society of North Aceh, Indonesia. They showed that several factors affect dermal problems, including poor environmental hygiene, wet and marshy conditions, densely populated living areas, and the habit of wearing tight, damp clothes. Furthermore, they found that 22.1% of fishermen suffered from tinea versicolor [38]. Based on these findings, it is reasonable to assign this increased prevalence among males to their possibly higher instance of outdoor activities, when compared to females. Conversely, males were found to be more resilient than females regarding certain digestive problems, such as ulcers (OR: 0.36). This trend is consistent with a previous study conducted with respondents age ranged between 20 and 40 years old in Surabaya City, Indonesia, in which a high prevalence of female respondents (77.8%) had ulcer problems [39]. In fact, [40] reported that the nationwide incidence of these health problems was as high as one case per thousand inhabitants.

Moreover, economic conditions were identified as the most significant factor related to the reported health problems. By factoring the high-income group in as a reference, we found that the low-income group was more vulnerable to certain problems related to mental health, bodily pain, digestive tracts, and respiratory health. Additionally, when compared to the high-income group, fidgeting and suffering from skin rashes were identified as the most common health problems among low-income and low-to-middle-income groups, respectively. These results suggest that the lower-income group may experience exposure to grueling environments in terms of economic, physical, and mental stress more frequently than other income groups. A study in India showed that the rates of mental health problems were twice as high in low-income than high-income groups [41]. This is compounded by the fact that low-income groups are often associated with poverty, crime, and are unable to access adequate living spaces or food since these factors can substantially increase their risk of mental health problems [42].

**Table 4. Results of logistic regression models and odd ratios along with the 95% confidence intervals for the covariates related to dwelling locations.**

| Dwelling location | Health problems | Estimate | p-value | SE | OR | OR with 95% CI | |
|---|---|---|---|---|---|---|---|
| | | | | | | Lower limit | Upper limit |
| *Type of district (urban as a control group)* | | | | | | | |
| Rural | Chest pain | 1.06 | 0.00873 ** | 0.40 | 2.88 | 1.34 | 6.60 |
| | Puke | 0.94 | 0.0106 * | 0.37 | 2.55 | 1.26 | 5.36 |
| Urban Kampung | Odynophagia | 1.64 | 0.0004 *** | 0.44 | 5.17 | 2.27 | 13.32 |
| | Diarrhoea | 1.37 | 0.00013 *** | 0.26 | 3.94 | 2.37 | 6.69 |
| | Headache | 1.29 | 0.0001 *** | 0.25 | 3.64 | 2.26 | 5.94 |
| | Fidgeting | 1.24 | 0.00002 *** | 0.28 | 3.47 | 2.01 | 6.13 |
| | Muscle cramps | 1.21 | 0.0001 *** | 0.24 | 3.35 | 2.10 | 5.42 |
| | Insomnia | 1.14 | 0.00002 *** | 0.27 | 3.14 | 1.87 | 5.37 |
| | Cough | 0.83 | 0.0003 *** | 0.24 | 2.30 | 1.44 | 3.69 |
| | Tachycardia | 0.83 | 0.025 * | 0.37 | 2.30 | 1.12 | 4.86 |
| | Rheumatism | 0.82 | 0.00396 ** | 0.28 | 2.27 | 1.31 | 4.00 |
| | Fatigue | 0.76 | 0.00209 ** | 0.25 | 2.13 | 1.32 | 3.47 |
| | Migraine | -0.64 | 0.0195 * | 0.27 | 0.53 | 0.3 | 0.90 |
| | Hypotension | -1.17 | 0.0009*** | 0.35 | 0.31 | 0.15 | 0.60 |
| | Sinusitis | -1.52 | 0.00255 ** | 0.50 | 0.22 | 0.07 | 0.54 |
| | Haemorrhoids | -1.64 | 0.0337 * | 0.77 | 0.19 | 0.03 | 0.73 |

*p < 0.05;

**p < 0.01;

***p < 0.001 by using Chi-squared test;

SE: Standard Error.

Tampubolon and Hanandita (2014) also confirmed the findings of studies like Lund et al. (2010) which indicate that higher levels of poverty are associated with more depressive symptoms [42, 43].

In addition to demographic variables, a similar logistic regression analysis was conducted with the variables related to dwelling conditions, including their locations (urban, rural, and urban kampung). A group of respondents living in urban areas were designated as a reference category for the covariates of the dwelling factors. Table 4 displays the results of this analysis. The difference between rural and urban areas, for most of the symptoms, was insignificant. However, respondents from an urban kampung exhibited significant differences from the urban respondents in a total of 14 symptoms; most of these differences were evaluated as positive estimates, namely higher possibilities of suffering from the symptoms. Health problems related to the digestive tract seemed to be the most common symptoms found in these districts; for example, both odynophagia and diarrhoea had high odds ratios. Poor sanitation, inadequate clean water, and unhygienic living spaces in the urban kampung might be the main causes of these common health problems [18]. In this context, this study is consistent with previous studies by Garg et al. (2020) and Cameron et al. (2021), who examined whether the causes of diarrheal diseases in Indonesia are associated with household conditions and water quality by using over 6000 data points from the Indonesia Family Life Survey (IFLS) [44]. Furthermore, rural areas in Indonesia generally also face insufficient infrastructure related to sanitation and water supply, when compared to urban areas. A previous study reported, based on nationwide data, that the prevalence of diarrhoea among children under two years old in rural areas (9.5%) was higher than in urban areas (7.7%) [45]. Therefore, further surveys are important to clearly determine the relationships between the higher

**Table 5. Results of logistic regression models and odds ratios along with the 95% confidence intervals (CI) for covariates related to occupants' indoor environments and air quality satisfaction.**

| Occupant evaluations | Health problems | Estimates | p-value | SE | OR | OR with 95% CI | |
|---|---|---|---|---|---|---|---|
| | | | | | | Lower limit | Upper limit |
| Being satisfied with indoor air quality | Cough | -0.77 | 0.0038 ** | 0.27 | 0.46 | 0.27 | 0.77 |
| | Puke | -0.83 | 0.0040 ** | 0.29 | 0.44 | 0.25 | 0.77 |
| | Dyspnea | -1.29 | 0.0008 *** | 0.39 | 0.27 | 0.13 | 0.59 |
| | Fainting | -1.60 | 0.0027 ** | 0.53 | 0.20 | 0.07 | 0.59 |
| Being satisfied with indoor environment | Hypotension | -0.71 | 0.0232 * | 0.31 | 0.49 | 0.27 | 0.91 |
| | Severe dehydration | -0.85 | 0.0052 ** | 0.31 | 0.43 | 0.23 | 0.78 |
| | Rheumatism | -1.03 | 0.0011 ** | 0.31 | 0.36 | 0.19 | 0.66 |
| | Tachycardia | -1.02 | 0.0080 ** | 0.38 | 0.36 | 0.17 | 0.77 |
| | Chest pain | -1.13 | 0.0054 ** | 0.41 | 0.32 | 0.07 | 0.82 |
| | Skin rash | -1.33 | 0.0007 *** | 0.39 | 0.27 | 0.12 | 0.57 |
| | Urinary hesitancy | -1.42 | 0.0232 * | 0.62 | 0.24 | 0.15 | 0.72 |

*$p < 0.05$;

**$p < 0.01$;

***$p < 0.001$ by using Chi-squared test;

SE: Standard Error.

prevalence of health problems in an urban kampung and other physical factors related to housing environments and urban infrastructure.

Table 5 summarizes the results of the logistic regression and OR analysis regarding the prevalence of health problems and the occupants' subjective evaluation of their indoor environments. Since the logistic regression deals with binary response variables, we simplified the occupants' responses to coincide with their levels of satisfaction (i.e., "satisfied" and "very satisfied") and dissatisfaction (i.e., "slightly dissatisfied", "dissatisfied"). The respondent groups with negative evaluations, or who expressed a dissatisfaction with the abovementioned conditions, were designated as references. For instance, the negative coefficient estimated for a cough can be interpreted as the odds of coughing occurring among respondents who were satisfied with their indoor air quality was 54% (1 –OR × 100%). These odds are lower in comparison to those who were not satisfied, while all other independent variables were kept at the same level. This suggests that when people have respiratory problems, they are more likely to be aware of air pollution. This is also consistent with a previous study which revealed that respiratory problems may be caused by the cumulative indoor air exposure to organic and inorganic air contaminants [46]. Additionally, we found that skin rashes, severe dehydration, urinary hesitancy, and hypotension were significantly correlated with the factors related to occupants' satisfaction with indoor environments with ORs between 0.24 and 0.49. These results exhibit a similar tendency to that of another cross-sectional study which included 242 respondents living in residential buildings in Jakarta [47]. They pointed out that eye irritations, skin rashes, headaches, fevers, and coughing bouts occurred in 3.2%, 12.9%, 12.9%, 32.2%, and 38.7% of the occupants who complained about their indoor environments, respectively. Moreover, being satisfied with their indoor environments was more prominently associated with the various health problems than being satisfied with air quality was. Furthermore, since the indoor environment satisfaction is a more comprehensive evaluation of the living environment (including air quality, visual, acoustic, and thermal comfort) this result is considered to be meaningful [48].

## Structural equation modelling

Considering the results displayed in Tables 3 and 4, we further hypothesized that economic conditions have significant direct and indirect effects on health problems, especially when considering the quality of living space. Hence, we utilized structural equation modelling (SEM) to accommodate exogenous variables, such as economic levels in a single model affecting health problems, with the perceived indoor environments and thermal discomfort experiences as mediators. With the 16 most common health problems, according to the logistic regression analysis described in Section 3, we modelled the conceptual direct and indirect effects of economic levels according to the weighting factors of the linear relations specified among latent variables (i.e. perceived indoor environments, thermal discomfort experiences, and health problems) using LISREL 8 [49]. Table 6 lists the designated independent variables of the SEM. The measured variable of economic levels represents the economic circumstances of the respondents; it consists of four categories scored from 0 (low), 1 (low-middle), 2 (middle-upper), and 3 (high). Additionally, the variables of the subjectively evaluated indoor air quality and indoor environment satisfaction were also scored from 0 to 3, with higher scores indicating better evaluations. Meanwhile, the measured variables pertaining to "struggling to sleep" and "hot indoor air exposure" were defined by the amount of times the respondents indicated they experienced the variables; they were divided into five categories scored from 0 (never) to 4 (every day or every night).

The quality of the analysis, as based on the structural model, was assessed with the standard model fit recommended by Yang et al. (2020)–who studied the modelling pathways between perceived residential green spaces and mental health. This standard model fit consists of a goodness-of-fit analysis, including a root mean square error of approximation (RMSEA < 0.06), Chi-squared value ($\chi^2$) divided by the degree of freedom (DF) ≤ 5, goodness-of-fit statistic (GFI > 0.95), adjusted goodness-of-fit statistic (AGFI > 0.95), normed-fit index (NFI ≥ 0.90), comparative fit index (CFI > 0.95), parsimony normed fit index p-value (PNFI > 0.50), parsimony goodness-of-fit index (PGFI > 0.50), and p < 0.05 [50].

Fig 5 illustrates the hypothetical model of SEM. It emphasizes a structural model in which the latent variable (health problems) is projected as a function of other variables, such as economic levels, perceived indoor environments, and thermal discomfort experiences. This model yielded acceptable good-fit data ($\chi^2$ = 247.06, DF = 166, RMSEA = 0.035, GFI = 0.95, AGFI = 0.93, NFI = 0.93, CFI = 0.98, PNFI = 0.74, PGFI = 0.68, p < 0.001). Accordingly, we identified that the economic level and thermal discomfort experiences of respondents were found to have significant direct effects on health problems with standardized estimation coefficients of -0.29, and 0.55, respectively (see Table 7). The loading factor from the economic level to health problems is predicted to exhibit an opposite correlation: the higher economic level of

**Table 6. Independent variables for structural equation modelling.**

| Variables | Categories | Variable values |
|---|---|---|
| Economic levels | 4 | 0 = low; 1 = lower-middle; 2 = middle-upper; 3 = high |
| Subjective indoor air quality evaluations | 4 | 0 = poor; 1 = slightly poor; 2 = good; 3 = very good |
| Indoor environment satisfaction | 4 | 0 = dissatisfied; 1 = slightly dissatisfied; 2 = satisfied; 3 = very satisfied |
| Struggling to sleep | 5 | 0 = Never; 1 = less frequent; 2 = a few nights per month; 3 = a few nights per week; 4 = every night |
| Hot indoor air exposure | 5 | 0 = Never; 1 = less frequent; 2 = a few days per month; 3 = a few days per week; 4 = everyday |

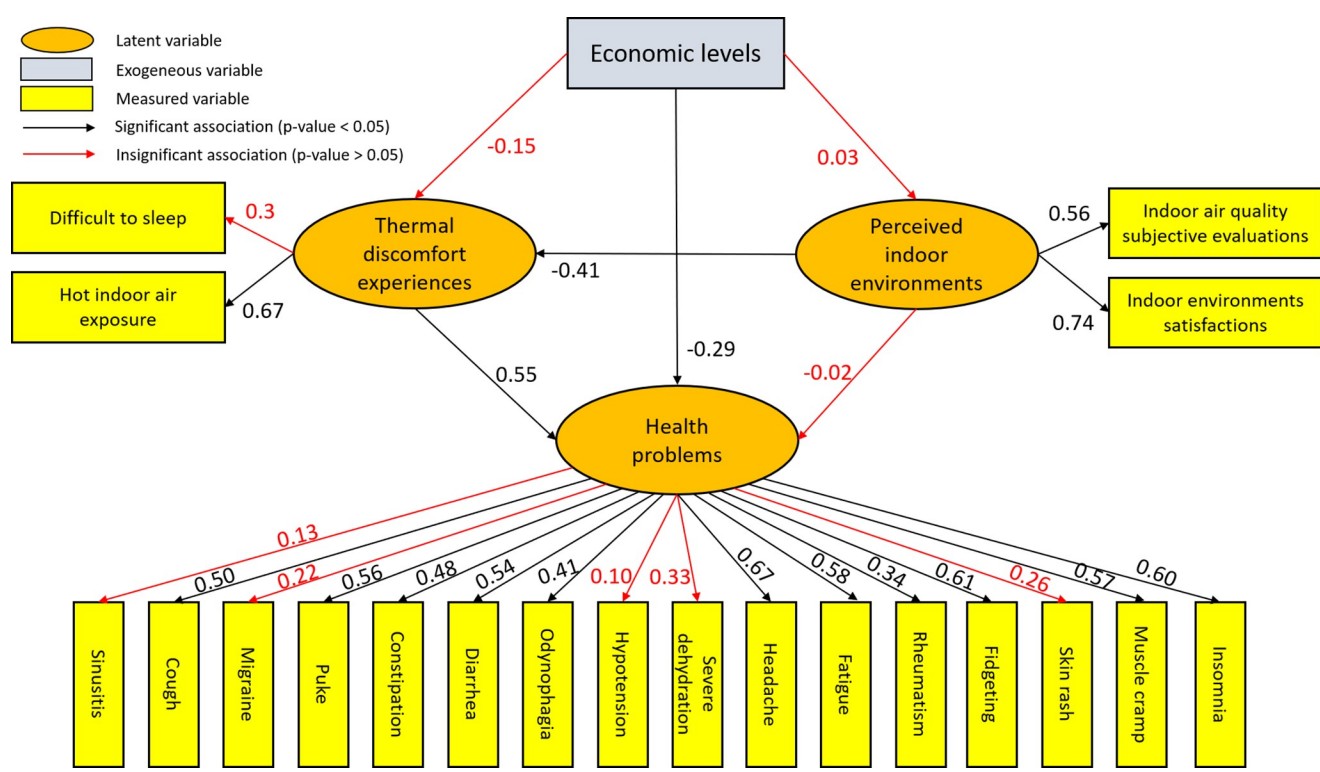

**Fig 5. Standardized estimates of the associations between economic levels, thermal discomfort experiences, perceived indoor environments, and health problems.**

the residents will result in lower weight factors for health problems (reflective construct variables). This is reflective of the results in a previous study conducted by Chen et al. (2019), who confirmed a stronger link between social economic status and health conditions, when compared to lifestyle and general indoor environments. Other findings identified that low economic conditions could lead to other factors, such as comorbidity, lifestyle, and unhygienic living environments, that might worsen the occupants' health conditions [51, 52]. Furthermore, thermal discomfort experiences exhibit a positive correlation with health problems; this signifies that respondents who struggle to sleep and experience hot indoor air exposures are predicted to be susceptible to certain health problems. Moreover, perceived indoor

**Table 7. The evaluations of path coefficients with respect to latent variables.**

| Paths | Estimate | SE | R² | Standardized estimates | p-value |
|---|---|---|---|---|---|
| EL → PIE | 0.02 | 0.95 | 0.0007 | 0.03 | 0.66 |
| EL → TDE | -0.12 | 0.81 | 0.19 | -0.15 | 0.053 |
| EL → HP | -0.25 | 0.1 | 0.45 | -0.29 | 0.02* |
| PIE → TDE | -0.41 | 0.15 | 0.19 | -0.41 | 0.007** |
| PIE → HP | -0.02 | 0.1 | 0.45 | -0.02 | 0.85 |
| TDE → HP | 0.55 | 0.27 | 0.45 | 0.55 | 0.04* |

Note: EL = Economic levels; PIE = Perceived indoor environments; TDE = Thermal discomfort experiences; HP = Health problems;

*p <0.05;

**p <0.01.

**Table 8. The loading factor results derived from latent variables of structural equation modelling.**

| Latent variables | Symptoms | Estimates | SE | $R^2$ | Standardized estimates | p-value |
|---|---|---|---|---|---|---|
| Health problems | Sinusitis | 0.04 | 0.09 | 0.02 | 0.13 | 0.14 |
| | Cough | 0.25 | 0.19 | 0.25 | 0.50 | 0.01* |
| | Migraine | 0.01 | 0.19 | 0.05 | 0.22 | 0.13 |
| | Puke | 0.23 | 0.12 | 0.31 | 0.56 | 0.02* |
| | Constipation | 0.20 | 0.14 | 0.23 | 0.48 | 0.01* |
| | Diarrhoea | 0.25 | 0.14 | 0.29 | 0.54 | 0.02* |
| | Odynophagia | 0.12 | 0.07 | 0.17 | 0.41 | 0.02* |
| | Hypotension | 0.04 | 0.17 | 0.01 | 0.10 | 0.11 |
| | Severe dehydration | 0.14 | 0.15 | 0.11 | 0.33 | 0.07 |
| | Headache | 0.34 | 0.14 | 0.45 | 0.67 | 0.01* |
| | Fatigue | 0.29 | 0.16 | 0.34 | 0.58 | 0.01* |
| | Rheumatism | 0.14 | 0.13 | 0.12 | 0.34 | 0.02* |
| | Fidgeting | 0.26 | 0.11 | 0.38 | 0.61 | 0.01* |
| | Skin rash | 0.08 | 0.09 | 0.07 | 0.26 | 0.08 |
| | Muscle cramp | 0.29 | 0.17 | 0.33 | 0.57 | 0.01* |
| | Insomnia | 0.26 | 0.12 | 0.36 | 0.60 | 0.01* |

*$p < 0.05$;

**$p < 0.01$;

***$p < 0.001$.

environments significantly affected the respondents' thermal discomfort experiences, with a negative coefficient of -0.41. This suggests that respondents with positive perceptions of indoor environments have lower risks of experiencing indoor thermal discomfort. Although the variable of perceived indoor environments had no direct impact on health problems, it was found to affect the respondents' health problems by influencing indoor thermal discomfort experiences.

Furthermore, Table 8 lists the loading factor results that were derived from the latent variables of the SEM. Among the statistically significant reflective constructs concerning health problems in this study, the highest contribution represented by standardized estimates was found for headaches (0.67), followed by fidgeting (0.61), insomnia (0.60), fatigue (0.58), muscle cramps (0.57), puking (0.56), diarrhoea (0.54), coughing (0.50), constipation (0.48), odynophagia (0.41), and rheumatism (0.34). It should be emphasized that these symptoms can be recognized as those that are more strongly affected by socioeconomic or environmental factors.

## Conclusion

The present study examined the influences of basic demographic factors and the subjective evaluation of indoor environments on respondents' health conditions through a questionnaire survey in Indonesia. Logistic regression and odds ratios were used to investigate the associations between the possibility of health problems occurring and certain personal factors, perceived indoor environments, and dwelling conditions (independent variables). Additionally, we applied a structural equation model to emphasize the inter-relationships between these independent variables and their direct and indirect effects on subjective indoor environment evaluations, thermal discomfort experiences, and health problems. The noteworthy results are as follows:

- Among the observed demographic factors, economic conditions were identified as the most significant factor associated with health problems. When compared to the high-income group, we found that the low-income group was more vulnerable to several health problems related to mental health, bodily pains, the digestive tract, and respiratory functions. Additionally, the reported health problems associated with the digestive tract, such as odynophagia and diarrhoea, were more common among respondents from the urban kampung than those who live in urban or rural districts.

- The hypothetical SEM of this study indicated that the economic conditions and thermal discomfort experiences of the respondents had significant direct effects on their health problems. Meanwhile, although perceived indoor environments were found to significantly affect respondents' thermal discomfort experiences, they were not direct predictors of their health problems; however, they could impact their health problems through a mediating role.

Through the questionnaire survey, this cross-sectional design study revealed an association between the prevalence of certain occupant's health problems and the personal and subjective living quality of their environments. Nevertheless, we must point out the limitations of this study in determining the health conditions of respondents. The acquired data concerning the prevalence of health problems are based on a questionnaire; therefore, a discrepancy between the self-reported health conditions and reality might exist. Hence, numerous samples based on a large-scale medical survey collaborated with local stakeholders are needed to provide more reliable and evidence-based analyses for future studies.

## Supporting information

**S1 Appendix. Questionnaire survey form.**
(PDF)

**S1 Dataset.**
(CSV)

## Acknowledgments

The authors are grateful for a support and assistance from Mrs. Dian Rahayu Perwita Sari, Department of Medical Laboratory Technology, Yogyakarta Health Polytechnic, Indonesian Ministry of Health to provide insightful knowledge of public health assessments.

## Author Contributions

**Conceptualization:** Solli Murtyas, Aya Hagishima.

**Data curation:** Solli Murtyas, Nishat T. Toosty.

**Investigation:** N. H. Kusumaningdyah.

**Methodology:** Solli Murtyas, Nishat T. Toosty.

**Supervision:** Aya Hagishima.

**Writing – original draft:** Solli Murtyas, Nishat T. Toosty.

**Writing – review & editing:** Solli Murtyas, Aya Hagishima.

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
