## [Decision Letter · Decision Letter 0]

29 Apr 2021

PONE-D-21-09960

Association between the prevalence of occupants’ health problems to their demographic and subjective indoor environment quality: A cross-sectional questionnaire survey study in Java Island, Indonesia

PLOS ONE

Dear Dr. Solli Murtyas,

Thank you for submitting your manuscript to PLOS ONE. After careful consideration, we feel that it has merit but does not fully meet PLOS ONE’s publication criteria as it currently stands. Therefore, we invite you to submit a revised version of the manuscript that addresses the points raised during the review process.

We look forward to receiving your revised manuscript.

Kind regards,

Shah Md Atiqul Haq

Academic Editor

PLOS ONE

Journal Requirements:

2. Thank you for including your ethics statement:  "All procedures performed in this work, analysing data of participants, were in accordance with the research ethical standards of Kyushu University, Japan.

All the data obtained from the survey regarding participants private profile was confidential and the consent was waived by the ethics committee. ".  

Please amend your current ethics statement to confirm that your named institutional review board or ethics committee specifically approved this study.

4. We note that Figure 2 in your submission contain map images which may be copyrighted. All PLOS content is published under the Creative Commons Attribution License (CC BY 4.0), which means that the manuscript, images, and Supporting Information files will be freely available online, and any third party is permitted to access, download, copy, distribute, and use these materials in any way, even commercially, with proper attribution. For these reasons, we cannot publish previously copyrighted maps or satellite images created using proprietary data, such as Google software (Google Maps, Street View, and Earth). For more information, see our copyright guidelines: http://journals.plos.org/plosone/s/licenses-and-copyright.

1.              You may seek permission from the original copyright holder of Figure 2 to publish the content specifically under the CC BY 4.0 license. 

5. We note that Figure 3 includes an image of a participant in the study.

Additional Editor Comments:

Dear Author,

Thank you for sending us this very important and innovative article.

I have received the reviewers' report and based on their advice, I suggest a minor revision.

Please follow the reviewers' comments and suggestions and submit the revised version.

I wish you the best of luck,

Reviewers' comments:

Reviewer's Responses to Questions

**Comments to the Author**

1. Is the manuscript technically sound, and do the data support the conclusions?

Reviewer #1: Yes

Reviewer #2: Yes

Reviewer #3: Yes

2. Has the statistical analysis been performed appropriately and rigorously? 

Reviewer #1: N/A

Reviewer #2: Yes

Reviewer #3: Yes

3. Have the authors made all data underlying the findings in their manuscript fully available?

Reviewer #1: Yes

Reviewer #2: Yes

Reviewer #3: Yes

4. Is the manuscript presented in an intelligible fashion and written in standard English?

Reviewer #1: Yes

Reviewer #2: Yes

Reviewer #3: Yes

5. Review Comments to the Author

Reviewer #1: Clarify this sentence in the abstract 'Additionally, perceived indoor environment quality could influence health problems by mediating thermal discomfort experiences.', mediating should be changed with more suitable word.

Line 99: It should be susceptible.

Line 182: People who expressed satisfaction were those who struggled to sleep or suffered indoor hear is in conflict. Please correct this sentence.

Line 188: spelling of emphasized

Line 197: Indoor and outdoor air quality of slums in Chile were worst than urban areas. How does this correlate to rural areas? Slums are entirely different than rural areas. Please clarify or remove this statement

In respiratory health conditions, authors mention 'blown' as a condition. What is it? I could only find on google that this term is used to emphasize disease severity. if it is a disease itself, please clarify in text or table.

Table 2: I guess authors meant 'Fidgeting and Fainting' instead of Fidgety and Fainted.

Same in rest of text. Please cross-check which word suits where

Do check if odds ratio is less than 1, it is negatively correlated instead of positive correlation

Authors should also mention or elaborate if there are some conditions that are genetic and may not be consequence of environmental conditions. In case of hypertension, it might be linked to both environment or genetics in a population.

Reviewer #2: understanding the basic relationship structure between the built environment, anthropometric factors, and economic conditions for a specific region is vital. Hence, this study intends to reveal the relationship between perceived indoor environments, experiences of indoor thermal discomfort, and occupant health conditions in Indonesia by means of a subjective questionnaire survey. Furthermore, the factors related to indoor environment quality (IEQ), economic levels, locations, and basic anthropometric information are also considered as potential explanatory variables for health condition. The manuscript is well written and addresses are the research questions. Therefore can be accepted in this current form.

Reviewer #3: The paper results and their details of contained have original work, but the under the ethical issue, of avoiding displaying the photo of people whos get the information of data and get their approval first. In my opinion, the title needs more concise. The supporting information, in particular, the maps, it needs professional maps and clear

6. PLOS authors have the option to publish the peer review history of their article (what does this mean?). If published, this will include your full peer review and any attached files.

Reviewer #1: No

Reviewer #2: No

Reviewer #3: No

---

## [Author Response · Author response to Decision Letter 0]

24 May 2021

We addressed the additional requirements based on the comments of editor as follows: 

1. We have checked the manuscript meets the style requirements of PLOS ONE. The main body and title, authors, and affiliations of the current manuscript have been checked to meet the PLOS ONE’s style requirements. 

2. We corrected the statements and included in Methods section of the manuscript as: 

“The questionnaire survey in this study was anonymous, and participants decided whether or not to participate after receiving sufficient explanation about the research objective. Obtaining a signed consent from participants were waived. All procedures performed in this study were in accordance with the ethical standards of Kyushu University and with the 1964 Helsinki declaration and its later amendments or comparable ethical standards, being approved by the Ethical Committee of Kyushu University.” (Line 114)

3. We decided to make the data available. The data, which are newly added in Appendix S1, includes preferences on indoor environment quality and self-reported health problems of 443 participants’ collected by the questionnaire. 

4. We replaced the previous map of Figure 2 with a free image provided at the below website. https://n.freemap.jp/tp/indonesia

In this website, the use of provided images have no restriction for publication. Although this website is written in Japanese, the website clearly explains their policy about free of charge for images at https://n.freemap.jp/st/price.html .

5. We removed the Figure 3 (the image of participants in the study) because we unable to obtain the consent form the subject of the photograph.

6. We have checked the reference list to ensure that it is complete and correct. 

Furthermore, we provided the responses to specific reviewer as follows: 

Reviewer1-#1 : According to your suggestion, we modified the sentence as: ‘Additionally, perceived indoor environment quality, which is possible to cause thermal discomfort experience, indirectly affect health problems’ (Line 29)

Reviewer1-#2: We changed the word as your suggested.

Reviewer1-#3: We corrected the sentence as: ‘Furthermore, based on respondents’ subjective evaluations, those who indicated that they were “very satisfied” with indoor environment conditions, generally show a small fraction of having difficulties to sleep in nighttime and suffered hot indoor air exposure.’

Reviewer1-#4: We changed the word as your suggested.

Reviewer1-#5: We decided to remove this statement as you suggested.

Reviewer1-#6: According to your suggestion, we replaced it with ‘Dyspnea’. It is the medical term for shortness of breath. 

Reviewer1-#7: Thank you for your advice. We replaced ‘fidgety’ and ‘fainted’ with ‘fidgeting’ and ‘fainting’ in the entire manuscript. 

Reviewer1-#8: We re-checked it and modified the text in line 220 as: ‘An OR less than 1 implies a negative relationship and vice versa’. Additionally, we modified the statements to be clearer in line 235 as follow: 

“a negative value of the estimated βj coefficients (OR<1) indicates a negative association between the prevalence of the health problem and the corresponding covariate. It means that respondents from a certain group of the corresponding covariate are less prone to the specific health problem compared to those in the reference category. Similarly, OR>1, which occurs for a positive value of βj estimates, indicates higher vulnerability of that specific group of the respondents to the health problem of interest than those of the reference group.”

Reviewer1-#9: Thank you very much for your insightful suggestion. 

As you suggested, it has been known that the interaction between genetic and environmental factors play an important role in hypertension based on past various medical studies. Considering the limited number of questions and participants of our survey, we think it is difficult to discuss about “whether there are some conditions that are genetic and may not be consequence of environmental conditions”. Although we cannot fully respond your suggestion, we modified the manuscript to be

“With regard to hypertension, it has been long known that interactions of multiple genetic and environmental factors play a significant role, and identified several indices associated with prevalence of hypertension such as body mass index (Pazoki et al., 2018; Peltzer and Pengpid, 2018; Zanchetti, 2016). Because of the limitation of sample number and type of questions, the survey cannot draw insight related to these factors, we can confirm that the prevalence of hypertension tends to be significantly correlated with the age group older than 50 years since its odds ratio is less than one; the lowest of all the age groups. Although the sample number of our survey is not quite large, this tendency is consistent with the Indonesian Ministry of Health’s report, which confirmed that 41% of adults older than 60 years old exhibited the hypertension in 2012, which is higher than the value of 17.1% for the age group of less than 25 years old (Giena et al., 2018; Widjaja et al., 2013).” (Line 247). 

References:

Giena, V.P., Thongpat, S., Nitirat, P., 2018. Predictors of health-promoting behaviour among older adults with hypertension in Indonesia. Int. J. Nurs. Sci. 5, 201–205. https://doi.org/10.1016/j.ijnss.2018.04.002

Pazoki, R., Dehghan, A., Evangelou, E., Warren, H., Gao, H., Caulfield, M., Elliott, P., Tzoulaki, I., 2018. Genetic predisposition to high blood pressure and lifestyle factors: Associations with midlife blood pressure levels and cardiovascular events. Circulation 137, 653–661. https://doi.org/10.1161/CIRCULATIONAHA.117.030898

Peltzer, K., Pengpid, S., 2018. The Prevalence and Social Determinants of Hypertension among Adults in Indonesia: A Cross-Sectional Population-Based National Survey. Int. J. Hypertens. 2018. https://doi.org/10.1155/2018/5610725

Widjaja, F.F., Santoso, L.A., Barus, N.R.V., Pradana, G.A., Estetika, C., 2013. Prehypertension and hypertension among young Indonesian adults at a primary health care in a rural area. Med. J. Indones. 22, 39–45. https://doi.org/10.13181/mji.v22i1.519

Zanchetti, A., 2016. Genetic and environmental factors in development of hypertension. J. Hypertens. 34, 2109–2110. https://doi.org/10.1097/HJH.0000000000001102

Reviewer2-#1: Thank you for giving us beneficial comments for our present work.

Reviewer3-#1: Thank you very much for your constructive insights. 

We modified the tittle more concise as you suggested. 

The updated title: Relation between occupants’ health problems, demographic and indoor environment subjective evaluations: A cross-sectional questionnaire survey study in Java Island, Indonesia 

Additionally, we have modified the supporting information. We deleted photograph of participants in the study because we unable to obtain the consent form the subject of the photograph.

---

## [Decision Letter · Decision Letter 1]

28 Jun 2021

Relation between occupants’ health problems, demographic and indoor environment subjective evaluations: A cross-sectional questionnaire survey study in Java Island, Indonesia

PONE-D-21-09960R1

Dear Dr. Solli,

We’re pleased to inform you that your manuscript has been judged scientifically suitable for publication and will be formally accepted for publication once it meets all outstanding technical requirements.

Kind regards,

Shah Md Atiqul Haq

Academic Editor

PLOS ONE

Additional Editor Comments (optional):

Reviewers' comments:

Reviewer's Responses to Questions

**Comments to the Author**

1. If the authors have adequately addressed your comments raised in a previous round of review and you feel that this manuscript is now acceptable for publication, you may indicate that here to bypass the “Comments to the Author” section, enter your conflict of interest statement in the “Confidential to Editor” section, and submit your "Accept" recommendation.

Reviewer #3: All comments have been addressed

2. Is the manuscript technically sound, and do the data support the conclusions?

Reviewer #3: Yes

3. Has the statistical analysis been performed appropriately and rigorously? 

Reviewer #3: Yes

4. Have the authors made all data underlying the findings in their manuscript fully available?

Reviewer #3: Yes

5. Is the manuscript presented in an intelligible fashion and written in standard English?

Reviewer #3: Yes

6. Review Comments to the Author

Reviewer #3: The idea was wonderful and, the ways of displaying all data clearly and easy for the reader, and I think this paper ready for publication.

7. PLOS authors have the option to publish the peer review history of their article (what does this mean?). If published, this will include your full peer review and any attached files.

Reviewer #3: No

---

## [Editor Report · Acceptance letter]

1 Jul 2021

PONE-D-21-09960R1 

Relation between occupants’ health problems, demographic and indoor environment subjective evaluations: A cross-sectional questionnaire survey study in Java Island, Indonesia 

Dear Dr. Murtyas:

I'm pleased to inform you that your manuscript has been deemed suitable for publication in PLOS ONE. Congratulations! Your manuscript is now with our production department. 

Kind regards, 

on behalf of

Dr. Shah Md Atiqul Haq 

Academic Editor

PLOS ONE